# FastSLM: Hierarchical Frame Q-Former for Effective Speech Modality Adaptation

## Abstract

Recent advances in large language models (LLMs) have demonstrated human-expert-level capabilities, driving significant interest in their potential for achieving artificial general intelligence (AGI). In particular, there is growing momentum in adapting LLMs to various modalities—including vision, video, and speech—through the development of multimodal LLMs (MLLMs). However, existing speech-language models (SLMs) research has largely overlooked cost-effective adaptation strategies for leveraging LLMs in the speech domain. In this paper, we propose FastSLM, a lightweight yet efficient SLM designed for effective understanding and reasoning over long-form speech. To address the challenge of aligning high-frame speech features with LLM, we introduce the hierarchical frame querying transformer (HFQ-Former), which compresses frame-level speech features while capturing both local and global context. Furthermore, we present a novel three-stage training strategy that enhances generalization across a wide range of speech-related tasks. Experimental results demonstrate that FastSLM achieves competitive performance compared to existing state-of-the-art (SOTA) models, despite operating with significantly lower FLOPs and parameter counts, while representing speech with only 1.67 tokens per second. The source code and model checkpoints are available at https://anonymous.4open.science/r/FastALM-1D6B.

## 1 Introduction

Recently, large language models (LLMs) (Achiam et al., 2023; Grattafiori et al., 2024; Comanici et al., 2025; Yang et al., 2025) have demonstrated expert-human level performance in various tasks such as code generation, math, and reasoning (Hendrycks et al., 2020; Wang et al., 2024). Accordingly, in order to move toward the ultimate goal of Artificial General Intelligence (AGI), research on multimodal LLMs (MLLMs) that apply various modalities (vision, speech, video) to LLM is actively being conducted (Yin et al., 2024; Lyu et al., 2023). Among them, speech is one of the core means by which artificial intelligence (AI) communicates with users, so speech recognition and understanding are essential components for achieving AGI (Sakshi et al., 2024). Consequently, much research has been conducted on ALMs (Tang et al., 2024; Abouelenin et al., 2025; Chu et al., 2023; 2024; Goel et al., 2025) that adapt the speech modality to LLM.

Many studies have used frame-level features extracted from pre-trained speech encoders such as Whisper (Radford et al., 2023), CLAP (Elizalde et al., 2023), Conformer (Gulati et al., 2020), and Beats (Chen et al., 2023) as inputs to LLM for speech modality adaptation. This approach is highly effective because it leverages frame-level features from encoders trained on large-scale speech data (Tang et al., 2024; Chu et al., 2023; 2024; Goel et al., 2025). However, projecting hundreds to thousands of frame-level features extracted by the encoder into the LLM using only a multi-layer perceptron (MLP) leads to a significant increase in computational complexity during generation. As the key-value (KV) cache of the autoregressive LLM grows, generation latency also increases, making real-time responses impractical (Arif et al., 2025).

Recently, the importance of long-form speech reasoning has been emphasized in tasks such as speech summarization (SSUM) and spoken-query-based question answering (SQQA) (Ghosh et al., 2024; Kang & Roy, 2024). However, these tasks require processing long-form speech input, which signif-

icantly increases the computational complexity of LLM. To address this challenge, we investigate a cost-effective design and training strategy for an SLM.

In this paper, we propose FastSLM, a lightweight and efficient SLM for processing speech input. To achieve this, we introduce the **H**ierarchical **F**rame **Q**uerying Trans**former** (HFQ-Former), a novel module that compresses frame-level features extracted from a speech encoder into compact features. Specifically, HFQ-Former reduces the input length to approximately 1.67 tokens per second by processing 3,000-frame speech chunks. We also propose a three-stage training strategy to jointly optimize performance across speech-based multitasks, including automatic speech recognition (ASR), automatic speech translation (AST), SSUM, and SQQA. Experimental results show that FastSLM attains competitive or even superior performance to state-of-the-art (SOTA) models across multiple benchmarks, despite being trained on comparatively small-scale datasets and with substantially fewer FLOPs. Additionally, to the best of our knowledge, this work introduces the first open-source SLM that supports Korean alongside English, thereby broadening accessibility and applicability to bilingual environments.

The following is a summary of our main contribution:

- We propose FastSLM, a low-cost and high-efficiency SLM that enables fast and effective text generation from long-form speech inputs with low computational overhead.
- We introduce the HFQ-Former module, which efficiently compresses frame-level features extracted from a pre-trained speech encoder.
- We present a three-stage training strategy that effectively adapts a pre-trained LLM for the speech modality, achieving strong alignment between speech and text representations without costly end-to-end training.
- Through extensive experiments on speech-based multitask evaluation, we demonstrate that FastSLM achieves competitive performance compared to existing SLM, while significantly reducing memory usage.
- We release the first open-source bilingual (Korean, English) multi-task SLM, providing a powerful and accessible foundation for future research in multilingual speech-language understanding.

## 2 RELATED WORK

### 2.1 AUDIO AND SPEECH LANGUAGE MODELS

Although LLMs have achieved human-expert capabilities, their generation is inherently grounded in textual input. Consequently, to advance toward AGI, numerous studies have explored extending LLMs to the audio (including speech) modality. Models such as AudioPaLM (Rubenstein et al., 2023), Kimi-Audio (Ding et al., 2025), Qwen-Audio (Chu et al., 2023), and Qwen2-Audio (Chu et al., 2024) demonstrate that frame-level speech features extracted from a speech encoder can be integrated into the LLM embedding space for end-to-end spoken understanding.

However, most of these models are primarily trained on short-form speech (typically under 30–60 seconds), limiting their ability to accurately process multi-minute inputs. To address this limitation, Voxtral-Mini, and Voxtral-Small (Liu et al., 2025a), Audio-Flamingo3 (Goel et al., 2025) introduced additional long-context training strategies and frame-level cross-attention for modality adaptation, enabling LLMs to better handle long-form speech. These works demonstrate that long-form speech reasoning is feasible when trained with sufficient data and computation.

Phi-4-Multimodal (Abouelenin et al., 2025), Gemini 2.5 (Comanici et al., 2025), and Qwen2.5-Omni (Xu et al., 2025) further show that LLMs can perform well across diverse modalities—including images, video, and speech—when trained on massive curated datasets. While powerful, their processing pipelines still depend on dense frame-level representations and incur substantial computational overhead for long-form speech inputs.

Notably, speech is one of the richest and semantically structured acoustic modalities. As a result, effective speech–LLM integration is essential for building systems capable of natural, interactive human communication. Despite strong progress, existing approaches typically rely on dense frame-

level cross-attention or repeated window-level processing, both of which introduce substantial computational cost when handling multi-minute speech. Even models explicitly trained on long-form speech do not directly address the challenge of efficiently aligning long-form speech representations with the LLM under strict FLOPs constraints.

This gap motivates our work. Rather than merely enabling LLMs to process long-form speech, we aim to design a mechanism that efficiently compresses and aligns long-range speech features.

### 2.2 Q-FORMER FOR SPEECH MODALITY ADAPTATION

To align speech features extracted from a speech encoder with a LLM, several prior works have explored using a multi-layer perceptron (MLP) to project speech features into the LLM embedding space or inserting cross-attention layers directly into the LLM Transformer blocks. Although these approaches enable effective multimodal fusion, they require architectural modifications to the LLM and substantially increase parameter size and computational cost.

To overcome these limitations, recent studies have introduced Q-Former–based compression modules that convert high-frame-rate speech features into a small set of learnable query tokens. SALMONN (Tang et al., 2024) proposed a window-level Q-Former that compresses speech within fixed temporal windows by attending over frame-level features extracted from Whisper and CLAP encoder. Segment-level Q-Former (Yu et al., 2024) instead partitions the speech sequence into length-based segments, applies Q-Former compression to each segment, and concatenates the resulting tokens before feeding them to the LLM.

Beyond these early designs, MMCE-QFormer (Xue et al., 2024) introduced a multimodal context-enhanced Q-Former to jointly fuse speech and textual cues for decoder-based LLMs. CompressedToFindLM (Liu et al., 2025b) proposed a reference-guided compression strategy, where each compressed token is generated from local frame-level features using learned prototype representations. AlignFormer (Fan et al., 2025) addressed the temporal mismatch between speech and text by integrating a CTC layer with a dynamic-window Q-Former, providing better alignment for autoregressive decoding.

## 3 METHODOLOGY

In Section 3.1, we describe the architecture of FastSLM and the inference process. In Section 3.2, we describe the training strategy employed for speech modality adaptation.

### 3.1 MODEL ARCHITECTURE OF FASTSLM

**FastSLM**: The overall architecture of the proposed FastSLM is illustrated in Fig. 1. FastSLM takes both speech and a text prompt as input to perform text generation. The processing pipeline is as follows. The raw waveform is first converted into a Mel spectrogram using a 25-ms window and a 10-ms stride. The resulting Mel spectrogram is then processed by a speech encoder (Radford et al., 2023) to extract frame-level features at a rate of 50 tokens per second. To use these frame-level features as input to the LLM, they are compressed and temporally aligned into 1.67 tokens per second using the proposed **HFQ-Former**. Finally, the speech tokens produced by the HFQ-Former are concatenated with the text tokens and provided as input to the LLM. Using this multimodal input, the model generates the final textual response.

Existing Q-Former (Li et al., 2023) is used to generate compact speech representations for the LLM by interacting with features $\hat{\mathbf{X}}^A$ extracted by the encoder from the input Mel spectrogram $\mathbf{X}^A$ (Tang et al., 2024). In this setup, learnable queries are refined through cross-attention with frame-level features extracted by the speech encoder (Yu et al., 2024). However, relying solely on the speech encoder output $\hat{\mathbf{X}}^A$ to learn queries limits the ability to capture the overall context of long-form speech. Furthermore, although segmenting speech features into windowed segments and processing them repeatedly helps to capture local information in $\mathbf{Q}^A$, it leads to a rapid increase in computational cost. To address these issues, we propose the HFQ-Former. The structure of HFQ-Former is illustrated in Fig. 2.

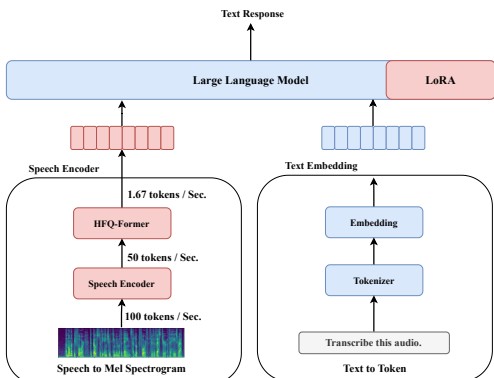

Figure 1: Architecture of FastSLM.

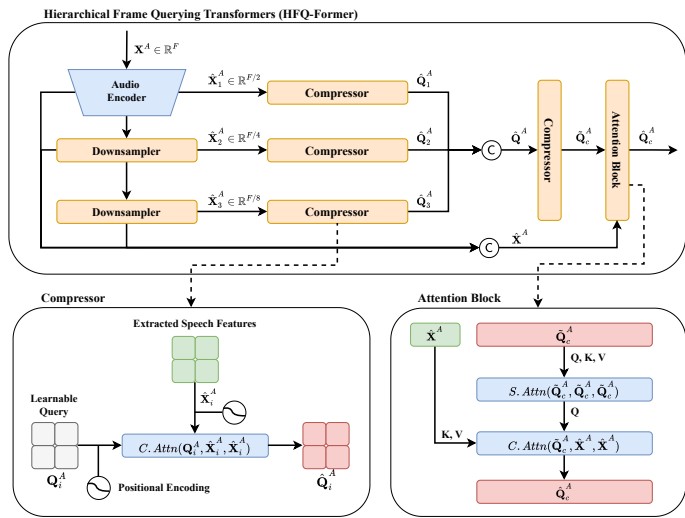

Figure 2: The proposed flowchart of HFQ-Former. where, C denotes the concatenation.

**HFQ-Former**: HFQ-Former adopts a three-stage hierarchical architecture (short-term, mid-term, long-term) to effectively align the high-frame speech features $\hat{\mathbf{X}}^A$ with the LLM. In **Stage 1**, the high-frame speech features $\hat{\mathbf{X}}_1^A$ retain rich local information and fused with the learnable query $\mathbf{Q}_1^A$ via cross-attention, producing the intermediate features $\hat{\mathbf{Q}}$. The learnable query $\mathbf{Q}_i^A$ is used as the query vector, while the speech feature $\hat{\mathbf{X}}_i^A$ serves as both the key and value. To preserve the positional information of the speech during compression, we incorporate positional encoding $PE(\cdot)$ (He et al., 2022) into the query and key features (Zhang et al., 2025). The cross-attention between $\mathbf{Q}_i^A$ and $\hat{\mathbf{X}}_i^A$ is then performed as follows:

$$\text{Compressor}(\mathbf{Q}_i^A, \hat{\mathbf{X}}_i^A, \hat{\mathbf{X}}_i^A) = \text{Softmax}\left( \frac{(\mathbf{Q}_i^A + PE(\mathbf{Q}_i^A)) \cdot (\hat{\mathbf{X}}_i^A + PE(\hat{\mathbf{X}}_i^A))^T}{\sqrt{d}} \right) \cdot \hat{\mathbf{X}}_i^A, \quad (1)$$

where $d$ denotes the attention dimensionality. In **Stage 2** and **Stage 3**, We perform cross-attention between the learnable query and the speech features. To capture broader temporal context in the speech, we down-sample the high-frame speech features using a Downsampler. Each Downsampler consists of two convolutional layers with kernel size 3 and a GELU activation function (Hendrycks & Gimpel, 2016), where the second convolution uses a stride of 2. This design enables the model to effectively capture both local and global temporal information from speech features.

At each $i$-th stage, the learnable query $\mathbf{Q}_i^A$ attends to the down-sampled speech features $\hat{\mathbf{X}}_i^A$ via cross-attention, enabling the model to integrate features at multiple temporal resolutions. The features at each stage can be written as follows:

$$\hat{\mathbf{Q}}_i^A = \text{Compressor}_i(\mathbf{Q}_i^A, \hat{\mathbf{X}}_i^A, \hat{\mathbf{X}}_i^A), \quad \text{where} \quad \hat{\mathbf{X}}_i^A = \text{Downsampler}_i(\hat{\mathbf{X}}_{i-1}^A), \tag{2}$$

where $i \in \{1, 2, 3\}$ and $\hat{\mathbf{X}}_0^A$ is the extracted high-frame speech features from speech encoder, and C.Attn denotes the cross-attention layer. At each stage, the generated $\hat{\mathbf{Q}}_i^A$ selectively extracts important information from the speech sequence, and the final representation is obtained by concatenating them as $\hat{\mathbf{Q}}^A = [\hat{\mathbf{Q}}_1^A; \hat{\mathbf{Q}}_2^A; \hat{\mathbf{Q}}_3^A]$. However, when considering long-form speech, the context size of $\hat{\mathbf{Q}}^A$ can still be computationally prohibitive for direct use as input to the LLM. We draw inspiration from vision-language models like LLaVA-mini (Zhang et al., 2025), which demonstrated that an entire image can be effectively represented by a single token. However, unlike a static image, speech is a temporal stream with complex, overlapping acoustic events, making extreme compression to a single token challenging without significant information loss. Therefore, to balance representational fidelity with computational efficiency, we compress hierarchical information $\hat{\mathbf{Q}}^A$ into a small fixed number of learnable queries $\mathbf{Q}_c^A$. The output learnable query $\tilde{\mathbf{Q}}_c^A$ is computed as:

$$\tilde{\mathbf{Q}}_c^A = \text{C.Attn}\left(\mathbf{Q}_c^A, \hat{\mathbf{Q}}^A, \hat{\mathbf{Q}}^A\right), \tag{3}$$

Subsequently, to capture salient features from the compressed speech tokens $\tilde{\mathbf{Q}}_c^A$ and to reference hierarchical information across diverse frames, we design an attention block that performs self-attention on $\tilde{\mathbf{Q}}_c^A$ followed by cross-attention with $\hat{\mathbf{X}}^A$. This cross-attention step is crucial as explicitly compensates for potential information loss caused by the dramatic compression of speech features. The output of this attention block, denoted as $\hat{\mathbf{Q}}_c^A$, is computed as:

$$\hat{\mathbf{Q}}_c^A = \text{Attention Block}\left(\tilde{\mathbf{Q}}_c^A, \hat{\mathbf{X}}^A\right), \tag{4}$$

where, $\hat{\mathbf{X}}^A = [\hat{\mathbf{X}}_0^A; \hat{\mathbf{X}}_1^A; \hat{\mathbf{X}}_2^A]$ denotes the concatenated frame-level speech features extracted at each stage. This produces the final compressed representation $\hat{\mathbf{Q}}_c^A$, which integrates both local and global context while reducing sequence length, thereby lowering the computational cost of autoregressive decoding in the LLM.

To further validate the effectiveness of the hierarchical compression, we provide a qualitative analysis of the cross-attention patterns across different stages in Appendix A. The visualization shows that HFQ-Former progressively shifts its attention toward deeper stages when processing long-form speech, confirming that the hierarchical design is essential for long-range temporal abstraction. To further validate the effectiveness of the hierarchical compression, we provide a qualitative analysis of the cross-attention patterns across different stages in Appendix B.

## 3.2 THREE-STAGE TRAINING STRATEGY

To train FastSLM, we propose a three-stage speech modality adaptation strategy, designed to progressively enhance the model capability to understand and adapt to speech input in LLM. Across all stages, we adopt low-rank adaptation (LoRA) (Hu et al., 2022) to ensure cost-efficient training with minimal trainable parameters. Specifically, we set the LoRA hyperparameter to a rank of 16 and an alpha of 64, resulting in a scaling factor of 4.

**Pre-training (Short-form speech Adaptation)**: In the first stage, the model is trained to adaptation short-form speech inputs. We construct a dataset of approximately 15K hours speech-text pairs in both Korean and English, with each speech clip restricted to under 30 seconds. This ensures that the model can learn general ASR capabilities and effectively align speech with language. We adopt prompt formats inspired by hierarchical tags (Chu et al., 2023; 2024) to improve language-specific understanding and detection during this stage. A detailed description of this can be found in Appendix C.

**Long-form Speech Adaptation**: Pre-trained speech encoders (Gulati et al., 2020; Radford et al., 2023; Elizalde et al., 2023; Chen et al., 2023) are typically limited to processing speech segments shorter than 30 seconds, which hinders their performance on long-speech tasks such as SSUM, SQQA. To address this limitation, Audio-Flamingo3 (Goel et al., 2025) constructed instruction tuning datasets specifically designed for long-form speech and audio understanding. However, building such datasets requires significant time and cost.

To provide a more cost-effective alternative, we train the model on a curated ASR-based dataset containing speech–text pairs of 1 to 15 minutes in length. This stage is designed not to training abstract reasoning directly but to strengthen the model's fundamental ability to process extended speech sequences. Through long-form transcription training, the model learns to maintain temporal coherence and preserve acoustic features over lengthy contexts—a critical prerequisite for complex downstream tasks. The resulting long-context representations supply the language model backbone with higher-quality, more coherent inputs, enabling superior performance on tasks such as SSUM and SQQA that require accurate understanding of an entire speech stream. Because ASR datasets are far more accessible than bespoke instruction-tuning dataset, this approach provides a practical and scalable path toward long-form speech modeling.

**Instruction Tuning**: In the final stage, we perform instruction tuning to enable the model to handle a variety of downstream tasks. Due to the scarcity of multi-task speech-language datasets in non-English languages, we generate a Korean multi-task dataset using the text-to-speech (TTS) engine (Zhao et al., 2023a), covering a range of tasks including SSUM, and SQQA. Unlike the previous stages, hierarchical tags are no longer required, as language identification capabilities have already been sufficiently established. However, hierarchical task tags are still employed to explicitly specify the task. A detailed description of this can be found in Appendix C.

Through this three-stage adaptation process, FastSLM is trained to achieve balanced capabilities across speech modality adaptation, long-form speech comprehension, and multi-task speech language understanding.

# 4 EXPERIMENT RESULTS

## 4.1 DATASET DESCRIPTION

**Pre-training Dataset**: As described in Section 3.2, we constructed a bilingual dataset comprising 15K speech-text pairs to adapt ASR capabilities to the LLM during the pre-training stage. Including 9,152 hours of English speech-text pair (LibriSpeech (Panayotov et al., 2015), GigaSpeech-L (Chen et al., 2021), Voxpopuli (Wang et al., 2021), SpgiSpeech-M (O'Neill et al., 2021), Earnings-22 (Rio et al., 2022), AMI (Kraaij et al., 2005), Common Voice 15 (Ardila et al., 2019), AI-HUB ASR-En (The Open AI Dataset Project, 2021)) dataset, and 7,812 hours of Korean speech-text pair (AI-Hub-ASR-Ko (The Open AI Dataset Project, 2021)) dataset. A detailed description of the pre-training dataset can be found in Appendix D Table 7.

**Long-form speech Dataset**: To enhance the model capacity to process long-form speech input, particularly for tasks such as SSUM and SQQA, we constructed a dedicated long-form speech dataset. For English, we curated a total of 219 hours of long-form speech. For Korean, we curated a total of 384 hours of long-form speech.

**Instruction Tuning Dataset**: To enable robust instruction-following capabilities, we constructed a multi-task instruction tuning dataset covering four representative speech-language tasks: ASR, AST, SSUM, and SQQA. For ASR, we randomly sampled Korean and English speech-text pairs from the during pre-training. For SSUM, our dataset includes 1,600 hours of long-form dialogue from the MNSC corpus (Wang et al., 2025). As no public Korean SSUM dataset was available, we synthesized one by applying a TTS engine to the KMSS text summarization dataset (Kim et al., 2022). For SQQA, we used the English LibriSQA dataset (Zhao et al., 2024). For Korean, we constructed a parallel dataset by converting the text-based KorQuAD dataset (Lim et al., 2019) into speech via TTS. A detailed breakdown of datasets used for each instruction tuning task is provided in Table 1.

**Evaluation Datasets**: To evaluate the speech understanding capabilities of **FastSLM**, we conducted experiments across a variety of benchmark tasks.

Table 1: Details of the instruction tuning dataset. "En" denotes English, "Ko" denotes Korean, and "En2Ko", "Ko2En" indicate the translation directions.

| Task | Dataset | Duration (hours) | #Samples | speech Language |
|------|---------|-----------------|----------|-----------------|
| ASR | LibriSpeech | 960 | 281,241 | En |
| | GigaSpeech-S | 250 | 230,068 | En |
| | AI-HUB ASR | 1500 | 320,000 | Ko |
| AST | AI-HUB AST (En2Ko) | 1,209 | 400,000 | En |
| | AI-HUB AST (Ko2En) | 1,152 | 400,000 | Ko |
| SSUM | SDS-PART6 | 1,600 | 103,935 | En |
| | KMSS | 668 | 84,000 | Ko |
| SQQA | LibriSQA | 364 | 104,014 | En |
| | KorQuAD-speech | 483 | 100,243 | Ko |
| Total | - | 8,186 | 2,023,501 | - |

- **ASR**: For English, we used the OpenASR evaluation datasets (Srivastav et al., 2023). For Korean, we used the Common Voice 15 (Ardila et al., 2019) and Fleurs (Conneau et al., 2022) datasets, which are open datasets, for fair comparison of results. We evaluate transcription quality using character error rate (CER) for Korean and word error rate (WER) for English to reflect the linguistic characteristics of each language.

- **AST**: We evaluated En2Ko and Ko2En translation on the Fleurs, and Minds14 (Gerz et al., 2021) dataset. In addition to the standard BLEU score (Post, 2018), we employed a GPT-4 based evaluation to overcome BLEU limitations with semantically valid but lexically different translations (Zhao et al., 2023b). Please refer to Appendix E for the AST judge prompt for GPT-4.

- **SSUM**: Evaluation was conducted on SDS-PART6 and KMSS-speech. Summarization quality was assessed using GPT-4 scoring with the LLM-as-a-judge framework (Zheng et al., 2023). Please refer to Appendix E for the SSUM judge prompt for GPT-4.

- **SQQA:** We measured accuracy on the LibriSQA and KorQuAD-speech datasets to evaluate SQQA performance.

## 4.2 EXPERIMENTAL SETUP

**Model Architecture**: FastSLM employs the encoder from Whisper-large-v3 (Radford et al., 2023) for speech feature extraction and adopts Qwen3-4B (Yang et al., 2025) as the backbone LLM for text generation. Despite its relatively compact size, Qwen3-4B exhibits sufficient capacity for comprehending speech-derived representations. In contrast to prior SLMs that typically utilize LLM backbone with 7 to 14 billion (B) parameters (Chu et al., 2023; 2024; Tang et al., 2024; Yu et al., 2024; Rubenstein et al., 2023; Ding et al., 2025; Liu et al., 2025a), FastSLM achieves a favorable cost-performance trade-off by leveraging lightweight architecture without compromising performance (Abouelenin et al., 2025; Ghosh et al., 2025). The HFQ-Former module within FastSLM compresses frame-level features via a hierarchical query-based mechanism. The number of learnable queries of $\mathbf{Q}_i^A$ was set to 80 cost-effectively (Yu et al., 2024), and the number of learnable queries of $\mathbf{Q}_c^A$ used as contextual input in LLM was set to 50 through the experiment. For further ablation studies and design justifications, please refer to Section 4.5 Fig. 3.

**Training**: FastSLM was trained on an NVIDIA A100 GPU-80GB×4 with a global batch size of 256. We used mixed precision training (Micikevicius et al., 2017) to maintain model performance while improving computational efficiency, with BF16 used as data type.

The model implementation details and the training setup are summarized in Appendix G.

## 4.3 COMPARISON WITH BASELINE Q-FORMER

To evaluate the performance of HFQ-Former, we compared it against two baseline methods: the segment-level Q-Former (SQ-Former) (Yu et al., 2024) and the window-level Q-Former (WQ-Former) (Tang et al., 2024), which explored speech modality adaptation using Q-Former. We additionally include an average pooling (AvgPool) baseline, in which frame-level speech features are downsampled through AvgPool and then projected into the LLM embedding space. This baseline

allows us to evaluate whether direct downsampling can effectively replace a learnable compression module.

For evaluation, we employed the ASR task, as it provides a direct measure of how accurately the model can understand speech content. In addition, to assess the computational load imposed on the LLM when processing long-form speech, we measured the FLOPs of the LLM using a 5-minute speech input. All baselines were trained and evaluated under the same pre-training dataset, LLM backbone, LoRA configuration, and embedding dimensions to ensure a fully fair comparison. The detailed results are presented in Table 2.

Table 2: Comparison of WER across baseline methods. LS denotes the LibriSpeech.

| Dataset Method | LS-clean | LS-other | Voxpopuli | #speech Tokens/Sec. | LLM FLOPs (T) |
|---|---|---|---|---|---|
| AvgPool | **1.88** | **4.12** | 7.14 | 25.0 | 30.6 |
| SQ-Former | 2.32 | 4.87 | 8.37 | 2.67 | 3.32 |
| WQ-Former | 2.14 | 4.51 | 7.26 | 2.93 | 3.65 |
| HFQ-Former (ours) | 2.09 | 4.67 | **6.99** | **1.67** | **2.51** |

As shown in Table 2, HFQ-Former achieves the best WER on VoxPopuli and remains highly competitive on LS-clean, while using the fewest speech tokens per second (1.67 tokens/sec). Although AvgPool attains slightly lower WER on the LibriSpeech benchmarks, it requires a vastly larger number of speech tokens (25 tokens/sec) and incurs a substantially higher LLM computation cost (30.6 TFLOPs).

In contrast, HFQ-Former achieves a strong balance between accuracy and efficiency: it reduces the token rate by 37% compared to WQ-Former and lowers LLM FLOPs by 31.2% (3.65T $\rightarrow$ 2.51T), while still improving WER on VoxPopuli. These results indicate that HFQ-Former provides a significantly more efficient speech-to-LLM alignment mechanism, greatly reducing the computational burden of autoregressive decoding for long-form speech without compromising recognition quality.

## 4.4 QUANTITATIVE RESULTS

We primarily compare FastSLM with strong speech-centric baselines such as WhisperV3 and AST, which directly align with our speech-only setting. For completeness, we additionally evaluate several multimodal models (Qwen2-Audio (Chu et al., 2024), Phi-4-Multimodal (MM) (Abouelenin et al., 2025), Gemini-2.5-Flash (Comanici et al., 2025), and Voxtral-Mini (Liu et al., 2025a)) in their speech-only mode. These models were not originally designed as SLMs, but we include them for an upper-bound comparison.

Table 3: Comparison of FastSLM with other SLMs on various tasks. WER is lower than best, accuracy (ACC), BLEU, and score higher than best. N/A indicates the model does not have such a capability. '*' indicates results fine-tuned on an additional Korean dataset provided in the official Hugging Face supplementary material. Detailed ASR benchmark WER for each dataset is reported in Appendix H.

| Task | Metric | Dataset | FastSLM 4.8B | Whisper 1.5B | Qwen2-Audio 8B | Phi4-MM 5.8B | Voxtral-mini 4.7B | Gemini-2.5-Flash |
|---|---|---|---|---|---|---|---|---|
| | #speech tokens/30 Sec. | | **50** | 1500 | 101 | 375 | 750 | 960 |
| ASR (En) | WER ↓ | OpenASR | 6.83 | 7.44 | 7.43 | **6.14** | 7.05 | 9.29 |
| ASR (Ko) | CER ↓ | Fleurs Common Voice 15 | **3.82** | 7.92 | N/A | N/A | N/A | 4.55 |
| AST (En2Ko) | BLEU/Score (1-5) ↑ | Fleurs | 7.20/3.94 | N/A | N/A | *2.62/2.69 | N/A | **13.4/4.56** |
| AST (Ko2En) | BLEU/Score (1-5) ↑ | Fleurs | 14.0/3.72 | 18.6/3.25 | N/A | *10.4/2.80 | N/A | **19.2/4.67** |
| AST (Ko2En) | BLEU/Score (1-5) ↑ | Minds14 | 26.3/4.12 | **29.5**/4.00 | N/A | *14.8/3.18 | N/A | 26.3/**4.52** |
| SSUM (En) | Score (1-7) ↑ | SDS-PART6 | 5.40 | N/A | 4.54 | 5.30 | 5.48 | **5.87** |
| SSUM (Ko) | Score (1-7) ↑ | KMSS | 4.12 | N/A | N/A | N/A | N/A | **4.37** |
| SQQA (En) | ACC ↑ | LibriSQA | **69.5** | N/A | 57.2 | 64.5 | 48.9 | 67.0 |
| SQQA (Ko) | ACC ↑ | KorQuAD-speech | **64.9** | N/A | N/A | N/A | N/A | 64.8 |

FastSLM demonstrates a powerful combination of efficiency and performance, achieving top-tier results with just 50 speech tokens per 30-second input—a fraction of that used by models like Whisper (1,500) and Gemini-2.5-Flash (960). As detailed in Table 3, its key achievements include:

- **ASR**: Achieves a SOTA CER of 3.82 on Korean benchmarks and a competitive WER of 6.83 on English OpenASR.
- **AST**: Showcases a preference for semantic quality over n-gram overlap, scoring higher in GPT-4 evaluation than Whisper on the Minds14 Ko2En task despite a lower BLEU score.
- **SSUM**: Delivers competitive scores of 5.40 (English) and 4.12 (Korean), performing on par with several larger models.
- **SQQA**: Sets the SOTA on all tested benchmarks, with leading accuracies of 69.5% on LibriSQA (English) and 64.9% on KorQuAD-speech (Korean).

In summary, FastSLM provides a highly effective and efficient solution for diverse speech-language tasks, proving that a compact speech representation can drive s SOTA performance.

## 4.5 ABLATION STUDY

**Effect of Hierarchical Staging in HFQ-Former:** To directly evaluate the benefit of the hierarchical design in HFQ-Former, we compare three variants using only Stage 1, Stage 1–2, and the full Stage 1–2–3. Table 4 shows that performance consistently improves as more hierarchical stages are included, demonstrating that progressive temporal abstraction is essential for long-form speech processing.

Table 4: Effect of hierarchical downsampling stages on speech understanding performance.

| Method \ Dataset | LS-clean WER ↓ | SDS-PART6-Speech Score (1-7) ↑ | KorQuAD-Speech ACC ↑ |
|---|---|---|---|
| Stage 1 | 2.23 | 4.12 | 56.7 |
| Stage 1/2 | 2.15 | 4.98 | 62.2 |
| Stage 1/2/3 | **2.09** | **5.40** | **64.9** |

**Effect of Speech Token Compression Ratio on ASR Performance**: To determine the optimal speech token compression ratio, we conducted an ablation study that evaluates the trade-off between ASR performance (WER) and computational cost. As illustrated in Fig. 3, a clear relationship emerges. While a high token rate (2.67 tokens/sec) yields the best ASR performance, it incurs a substantial computational cost. In contrast, an overly compressed representation (1.33 tokens/sec) results in a significant degradation of performance.

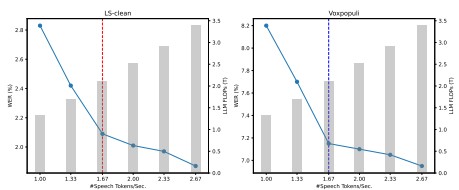

Figure 3: ASR performance of FastSLM with various speech tokens. (left) LS-clean decoding result, and (right) Voxpopuli decoding result.

Our analysis identifies 1.67 tokens/sec as an optimal operating point that balances these competing factors, achieving strong performance while minimizing computational demands. This decision is supported by a mathematical analysis of the point of diminishing returns, detailed in Appendix I.

**Scaling Limits of Long-Form Speech Input with FastSLM:** To provide a practical assessment beyond indirect complexity metrics like parameters and FLOPs (Ma et al., 2018), we empirically evaluate the scaling properties of FastSLM. We measure two key indicators on a single 40GB NVIDIA A100 GPU: VRAM consumption to assess memory efficiency and time-to-first-token (TTFT) to quantify the latency introduced by our multi-layered HFQ-Former. The results, presented in Fig. 4, highlight significant advantages in scalability. While benchmark models exhibit exponential VRAM growth, FastSLM demonstrates near-linear scaling, successfully processing an 8 hours (28,800 seconds) speech stream using under 30GB of memory. Regarding latency, FastSLM maintains a competitive TTFT. Although its initial latency is comparable to the similarly-sized Voxtral-Mini, it scales far more effectively, showing only a minimal increase as the speech length grows, in contrast to the sharp rise observed in the baseline. These findings confirm that FastSLM architecture enables robust inference on ultra-long-form speech far beyond the capabilities of existing models, especially within

constrained environments. Furthermore, its efficiency is highly advantageous for batch processing; the minimal memory footprint per stream allows for significantly larger batch sizes on a single GPU, thereby maximizing throughput for parallelized workloads.

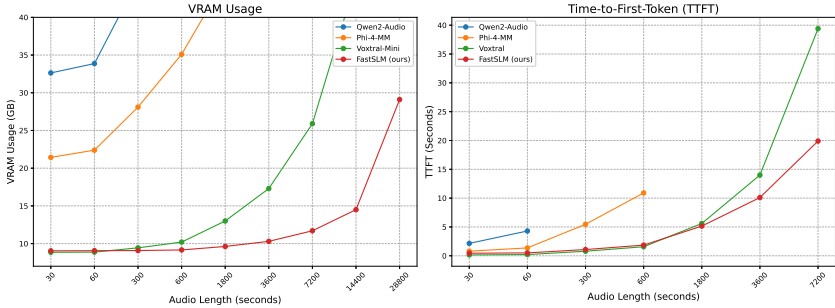

Figure 4: Comparison VRAM usage and time-to-first-token (TTFT) according to speech length.

**Effect of Hierarchical Modules and Training Stage:** To validate the effectiveness of our architecture and three-stage training strategy, we performed an ablation study by selectively removing key components: the Downsampler stage, the hierarchical attention mechanism, and Training Stage 2 (dedicated to long-form speech adaptation). The results, summarized in Table 5, highlight the critical contribution of each component. Removing the Downsampler stage or hierarchical attention leads to substantial performance degradation across all benchmarks—WER on LS-Long (Park et al., 2024) rises to 12.4 and 10.8, respectively. More importantly, omitting Training Stage 2, which enhances long-context understanding through ASR-based pretraining, results in notable drops across tasks, increasing LS-Long WER from 5.98 to 6.81 and reducing KorQuAD-Speech accuracy from 64.9% to 59.0%. The full FastSLM configuration consistently outperforms all ablated variants, demonstrating that both the hierarchical architectural design and the dedicated Training Stage 2 are indispensable for achieving efficient and robust long-form speech adaptation.

Table 5: Comparison of long-form speech adaptation strategy.

| Method \ Dataset | LS-Long WER ↓ | KorQuAD-Speech ACC ↑ | SDS-PART6 Score (1-7) ↑ |
|---|---|---|---|
| w/o Downsample Stage | 12.4 | 56.7 | 4.12 |
| w/o Hierarchical Attention | 10.8 | 56.9 | 4.92 |
| w/o Training Stage 2 | 6.81 | 59.0 | 5.07 |
| FastSLM | **5.98** | **64.9** | **5.40** |

## 5 CONCLUSION

In this paper, we introduce FastSLM, a lightweight and efficient SLM designed to overcome the critical scaling limitations of processing long-form speech. At the core of our approach is the HFQ-Former, a novel module that hierarchically compresses high-frame speech features into an optimal representation while preserving both local and global context. This architecture is complemented by a cost-effective three-stage training strategy, which enables robust adaptation of a pre-trained LLM for the speech modality. FastSLM achieves SOTA or competitive performance across diverse benchmarks while reducing the high-frame level features of conventional speech encoders by up to 97%, dramatically lowering GPU memory usage and computational cost without sacrificing accuracy. These results demonstrate that FastSLM provides a practical and scalable solution for efficient long-form speech understanding. By enabling models to process and reason over one of the primary modalities of human communication, this work contributes a critical building block for future multimodal systems aspiring toward AGI. Despite these promising results, we acknowledge several limitations of our current approach. A detailed discussion of these, along with potential directions for future research, is provided in Appendix J.

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

## A  QUALITATIVE ANALYSIS OF HFQ-FORMER ATTENTION MAP

Fig. 5 shows the cross-attention patterns of HFQ-Former for short-form (top), mid-form (middle), and long-form (bottom) speech across the three hierarchical stages.

For short-form speech, the final queries attend broadly to all stages, indicating that local and mid-term features remain useful when the sequence is short. For mid-form speech, attention begins to shift away from Stage 1 and is increasingly concentrated on Stage 2 and Stage 3, reflecting the need for broader temporal context. For long-form speech, attention becomes strongly dominated by Stage 3, while Stage 1 and Stage 2 receive minimal attention. This shows that the model relies on high-level, compressed representations for long-form speech reasoning.

Overall, as speech length increases, the attention distribution progressively moves from local (Stage 1) to global (Stage 3) features, demonstrating that HFQ-Former adaptively adjusts its focus based on speech length.

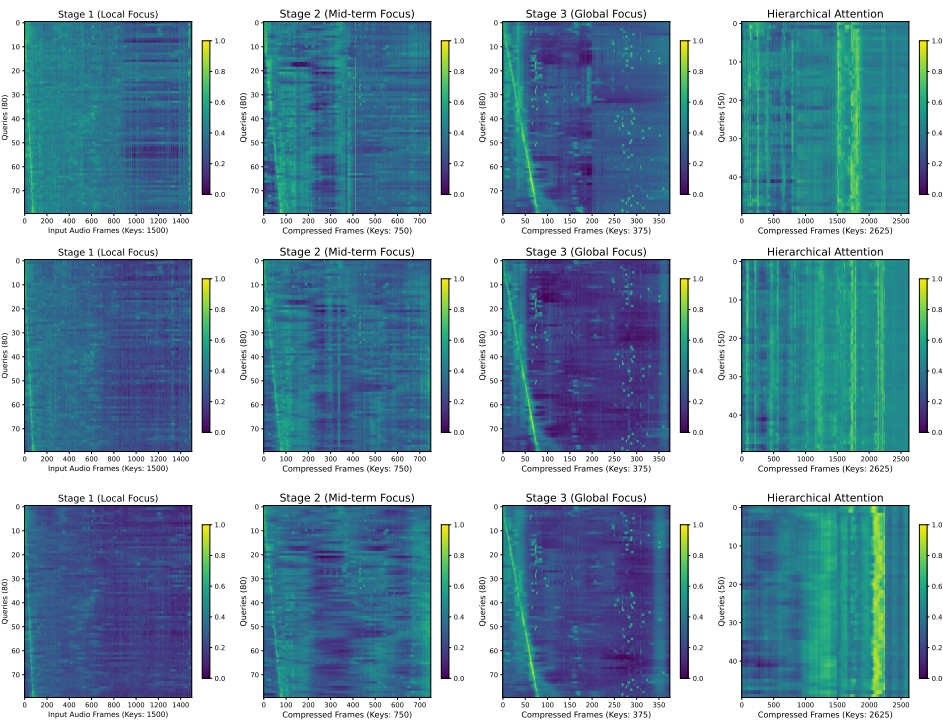

Figure 5: Visualization of HFQ-Former attention patterns across speech duration (logarithmic scale). Top: short-form speech ($< 30\,\text{s}$). Middle: mid-form speech ($< 60\,\text{s}$). Bottom: long-form speech ($> 15\,\text{min}$).

## B  ADDITIONAL ANALYSIS OF HIERARCHICAL TEMPORAL MODELING

To further substantiate the necessity of hierarchical temporal abstraction in HFQ-Former, we present additional analyses examining how different stages attend to both local and global speech context and compare HFQ-Former against a larger single-stage Q-Former operating under the same token budget. Local temporal cues correspond to short-range phonetic transitions, formant dynamics, and prosodic micro-patterns within 50 to 200 ms. In contrast, global temporal cues capture long-range structure such as discourse flow, topic progression, and speaker turns, which are especially important for long-form reasoning tasks including SSUM and SQQA. A single-stage Q-Former must simultaneously compress thousands of speech frames, forcing local and global cues to compete within a single attention scale, which leads to representational bottlenecks.

HFQ-Former alleviates this limitation through progressive temporal abstraction: early stages attend to fine-grained acoustic details, while deeper stages integrate increasingly broader temporal context. This hierarchical design enables efficient multi-scale modeling without increasing the token budget. Our cross-attention map visualizations (Appendix A) further confirm that attention shifts toward deeper stages as input duration increases, highlighting the importance of multi-stage processing for long-range temporal understanding. The empirical comparison between HFQ-Former and the larger single-stage Q-Former under the same token budget is presented in Table 6.

Table 6: Comparison between a larger single-stage Q-Former and our HFQ-Former under the same token budget.

| Method \ Dataset | LS-clean WER ↓ | Voxpopuli WER ↓ | SDS-PART6-Speech Score (1-7) ↑ | KorQuAD-Speech ACC ↑ |
|---|---|---|---|---|
| Larger Single-stage Q-Former | 2.11 | 7.11 | 5.04 | 61.2 |
| **HFQ-Former (ours)** | **2.09** | **6.99** | **5.40** | **64.9** |

Although the ASR quality (LS-clean) is comparable across the two models, HFQ-Former achieves substantially higher performance on long-form and reasoning-intensive tasks. These results empirically demonstrate that hierarchical temporal decomposition is critical for robust long-form speech understanding.

## C  PROMPT TEMPLATE AND HIERARCHICAL TAGS FOR TRAINING

We applied a unified prompt template and task/language control tokens during training to support multiple speech-language tasks:

User :< |audio_bos| >< |AUDIO| >< |audio_eos| > {Prompt/Question}\n Assistant :

To improve task specialization and language awareness, we employed hierarchical task and language tokens, which enabled robust detection and performance across languages and tasks:

Language Token :< |KO| >< |EN| >
Task Token :< |ASR| >< |AST| >< |SQQA| >< |SSUM| >

## D  PRE-TRAINING DATASET

Table 7 is a dataset used for pre-training of FastSLM. The dataset consists of English and Korean, and consists of a total of 10M speech-text pairs.

Table 7: Pre-training dataset details. En denote the English, and Ko denote the Korean.

| Dataset | Duration (hours) | # Samples | Speech Language |
|---|---|---|---|
| LibriSpeech | 960 | 281,241 | En |
| TED-LIUM-release3 | 454 | 268,263 | En |
| GigaSpeech-L | 2,500 | 2,266,371 | En |
| Voxpopuli | 523 | 182,482 | En |
| SpgiSpeech-M | 1,000 | 385,361 | En |
| Earnings-22 | 105 | 52,006 | En |
| AMI | 78 | 108,502 | En |
| Common Voice 15 | 2,532 | 1,070,066 | En |
| AI-HUB ASR-En | 1,000 | 1,020,265 | En |
| AI-HUB ASR-Ko | 7,812 | 4,557,512 | Ko |
| Total | 15,018 | 10,212,348 | - |

# E    PROMPT FOR GPT-4 AS A JUDGE ON SPEECH BENCHMARKS

The following is the exact prompt template used for evaluating AST output via an LLM-as-a-Judge. Placeholders like *SOURCE_TRANSCRIPT* are filled in programmatically for each evaluation instance.

Listing 1: LLM-as-a-Judge Prompt for AST Evaluation

```
# [Role]
You are a highly skilled professional evaluator specializing in {speech
    _language} to{target_language} Automatic Speech Translation (AST).
    Your task is to meticulously evaluate the quality of a {target_lang}
     translation generated from an English speech transcript.

Critically, you must recognize that the source text is not a manually
    written sentence but a transcript generated by an Automatic Speech
    Recognition (ASR) system. Therefore, the source itself may contain
    errors.

# [Evaluation Criteria]
You will conduct your evaluation based on the following two core
    criteria and an error analysis.

# 1. Adequacy: Accuracy of Meaning
Rate on a scale of 1-5 how accurately the core meaning and nuances of
    the source text are preserved in the target translation.
- **5 (Excellent):** All information and nuances from the source are
    perfectly conveyed without any loss or distortion.
- **4 (Good):** The core meaning is fully conveyed, but some minor
    nuances or details are lost.
- **3 (Fair):** The core meaning is conveyed, but some important
    information is missing or translated slightly inaccurately.
- **2 (Poor):** The subject, object, or key terms are mistranslated,
    leading to a significant distortion of the source's meaning.
- **1 (Very Poor):** The translation is a complete mistranslation and
    does not convey any of the source's meaning.

# 2. Fluency: Naturalness of the Translation
Rate on a scale of 1-5 how grammatically correct and natural the target
     translation sounds to a native Korean speaker.
- **5 (Excellent):** The translation is grammatically perfect and
    sounds completely natural, as if written by a native speaker.
- **4 (Good):** The translation is grammatically correct but feels
    slightly unnatural or "like a translation."
- **3 (Fair):** The meaning is understandable, but there are clear
    grammatical errors or awkward expressions.
- **2 (Poor):** The sentence structure is very awkward or contains many
     grammatical errors, making it difficult to understand.
- **1 (Very Poor):** The output is a collection of words that is not
    comprehensible as a sentence.

# [Critical Instructions for AST Evaluation]
- **Potential for ASR Errors:** If a word in the source transcript
    seems highly out of place in its context (e.g., the word 'bricks' in
     a description of a climbing destination), it is likely an ASR error
    .
- **Evaluating Error Correction:** If the translation model **corrects
    ** such a likely ASR error into a contextually appropriate term (e.g
    ., translating 'bricks' as the Korean term for 'granite walls', this
     is a highly positive attribute. You must mention this in the '
    adequacy_reason' and award a high score for Adequacy.

# [Inputs]
- **ASR Transcript (Source):** {SOURCE_TRANSCRIPT}
- **Model Translation (Candidate):** {MODEL_TRANSLATION}
```

```
- **Human Reference (Gold):** {HUMAN_REFERENCE}

# [Output Format]
You MUST provide your evaluation results in the following JSON format.

  {
   adequacy_score: <Float, 1.0-5.0>,
   fluency_score:  <Float, 1.0-5.0>,
   overall_score:  <Float, 1.0-5.0>
  }
```

Listing 2: LLM-as-a-Judge Prompt for SSUM Evaluation

```
You are a skilled evaluator for summaries generated based on user-
    provided instructions.
Your task is to rate how well the summary follows the user's
    instructions on a 1-7 scale.

Scoring Rubric:
- **7 (Excellent):** Fully follows all instructions. Accurate, fluent,
    and coherent with the correct level of detail and structure.
- **6 (Good):** Almost perfect, with very minor issues that do not
    affect usability (e.g., tiny structural deviation, trivial omission)
    .
- **5 (Mostly Correct):** Fulfills the main instruction but has
    noticeable issues (e.g., includes some unimportant extras, misses a
    few details).
- **4 (Acceptable):** Adheres to the instruction partially but has
    significant issues like inconsistencies or irrelevant content.
- **3 (Poor):** Minimally adheres to the instruction, missing most
    required details or containing significant irrelevant/hallucinated
    content.
- **2 (Very Poor):** Fails to follow the core instruction. Mostly
    irrelevant, fabricated, or ignores requested structure/tone.
- **1 (Fails):** Completely fails to follow instructions.

Input:
- **User Instruction:** {USER_INSTRUCTION}
- **Reference (gold):** {REFERENCE_ANSWER}
- **Model Summarization:** {SUMMARY_TO_EVALUATE}

Notes:
- It helps to read the Summary first, then compare with the Reference
    and Instruction.
- If the summary is missing or empty, return N/A as the score.

Output:
Note: Use the following JSON format for easy downstream consumption.
{
    explanation: "Brief reasoning for the score based on the rubric.",
    score: <Float, 1-7>
}
```

## F  GENERATION AND DECODING CONFIGURATION

We use the following decoding hyperparameters for all LLM-based generation tasks.

Table 8: Decoding configuration used for LLM-based generation in FastSLM.

| Parameter | Value |
|---|---|
| Decoding Strategy | Sampling |
| Temperature | 0.2 |
| Top-p | 0.95 |
| Top-k | 20 |
| Repetition Penalty | 1.0 |

## G  MODEL AND TRAINING PARAMETERS

The model implementation details and the training setup for each stage are presented in Table 9 and Table 10.

Table 9: Model configuration for FastSLM.

| Module | Component | Configuration |
|---|---|---|
| Encoder | Backbone | Whisper-large-v3 |
| | Parameters | 635M |
| | Hidden Size | 1280 |
| | Context Length | 1500 |
| Adapter | Backbone | HFQ-Former |
| | Parameters | 56M |
| | Hidden Size | 1280 |
| | Queries per Stage | 80 |
| | Compressed Speech Token | 50 |
| | Downsampling Factors | 2 |
| LLM | Backbone | Qwen3-4B |
| | Parameters | 4.06B |
| | Hidden Size | 2560 |
| | Context Length | 4096 |
| LoRA | Rank ($r$) | 16 |
| | Alpha ($\alpha$) | 64 |
| | Scaling Factor | 4 |
| | LoRA Target Modules | q/k/v_proj, gate/up/down_proj |

Table 10: Training settings across stages

| Setting | Stage1 | Stage2 | Stage3 |
|---|---|---|---|
| Learning Rate | 1e-4 | 5e-5 | 5e-5 |
| Learning Rate Scheduler | | Linear Decay | |
| Weight Decay | 0 | 1e-4 | 1e-4 |
| Epoch | 1 | 1 | 2 |
| Data Type | | BF16 | |
| DeepSpeed Stage | | Zero2 | |

## H  DETAILS OF ASR BENCHMARK RESULTS

Table 11 presents a detailed comparison of FastSLM and SOTA models across multiple ASR benchmarks. We report WER for English datasets and CER for Korean datasets. The results demonstrate that FastSLM achieves competitive performance while using significantly fewer speech tokens per second.

Table 11: Comparison of WER between FastSLM and state-of-the-art (SOTA) models. This results representation ASR Benchmark Dataset WER and CER.

| Dataset | Sub-Category | Metric | FastSLM 4.8B | Qwen2-Audio 8B | Phi4-Multimodal 5.8B | Whisper 1.5B | Voxtral-mini 4.7B | Gemini-2.5-Flash |
|---------|-------------|--------|--------------|----------------|----------------------|--------------|-------------------|------------------|
| OpenASR | AMI | WER | 12.8 | 15.2 | 11.7 | 16.0 | 16.3 | 21.6 |
|  | Earnings22 | WER | 10.5 | 14.1 | 10.2 | 11.3 | 10.7 | 13.1 |
|  | GigaSpeech | WER | 11.2 | 10.3 | 9.78 | 10.0 | 10.2 | 10.7 |
|  | SpgiSpeech | WER | 2.52 | 3.00 | 3.13 | 2.01 | 2.37 | 3.82 |
|  | TEDLIUM | WER | 3.84 | 4.05 | 2.90 | 3.91 | 3.68 | 3.01 |
|  | LS-clean | WER | 2.09 | 1.74 | 1.68 | 2.94 | 1.88 | 2.49 |
|  | LS-other | WER | 4.67 | 4.03 | 3.83 | 3.86 | 4.10 | 5.84 |
|  | Voxpopuli | WER | 6.99 | 7.05 | 5.91 | 9.54 | 7.14 | 7.89 |
| Fleurs | En | WER | 5.64 | 5.27 | 3.38 | 4.10 | 3.77 | 6.20 |
|  | Ko | CER | 2.79 | N/A | N/A | 5.32 | N/A | 3.00 |
| Common Voice 15 | En | WER | 12.0 | 8.68 | 7.61 | 9.30 | 10.2 | 11.2 |
|  | Ko | CER | 4.55 | N/A | N/A | 5.74 | N/A | 6.09 |

# I  MATHEMATICAL JUSTIFICATION FOR SPEECH TOKEN RATIO

To provide a more mathematical justification for our choice of 1.67 speech tokens/sec, we analyze the marginal gain in performance versus the marginal increase in cost, identifying the point of diminishing returns. We define a simple Efficiency Score to formalize this trade-off:

$$\text{Efficiency Score} = \frac{\Delta\text{Performance (WER Reduction)}}{\Delta\text{Cost (FLOPS Increase)}}$$

We apply this metric to the data from the LS-clean chart (Fig. 3). The results, summarized in Table 12, clearly show a sharp drop in efficiency after the 1.67 tokens/sec interval.

Table 12: Efficiency Score calculation for different token rate intervals on the LS-clean dataset. A higher score indicates greater efficiency.

| Token Rate Interval (tokens/sec) | $\Delta$WER (Reduction) | $\Delta$FLOPS (Increase) | Efficiency Score $\uparrow$ |
|----------------------------------|-------------------------|--------------------------|------------------------------|
| $1.33 \rightarrow 1.67$ | $\approx 0.3\%$ | $\approx 0.3\,\text{T}$ | $\approx \mathbf{1.00}$ |
| $1.67 \rightarrow 2.00$ | $\approx 0.1\%$ | $\approx 1.3\,\text{T}$ | $\approx \mathbf{0.08}$ |

The analysis in Table 12 quantitatively demonstrates that the interval beyond 1.67 tokens/sec marks a stark point of diminishing returns. Although adding more tokens continues to slightly lower the WER, the computational cost required for each marginal improvement becomes disproportionately high. Therefore, 1.67 tokens/sec is the most efficient configuration, maximizing the performance gain before the cost-benefit ratio sharply declines.

# J  LIMITATION

While FastSLM demonstrates a significant step forward in creating computationally efficient and scalable SLMs, this work has several limitations that warrant consideration and offer avenues for future research.

First, our approach to enabling Korean-language capabilities relies heavily on instruction-tuning datasets generated with a TTS engine, which introduces a potential synthetic-to-real domain gap. Synthetic speech generally lacks natural prosody, spontaneous disfluencies (e.g., hesitations, restarts), background noise, and speaker variability that occur in real conversational speech. As a result, although FastSLM shows strong performance on our synthetic Korean evaluation sets, its ability to generalize to authentic, real-world Korean speech remains unverified and may be significantly lower. Moreover, because evaluation is largely performed on data drawn from the same distribution as the training data, current metrics may overestimate real-world performance.

Second, FastSLM is evaluated exclusively on speech tasks. While this focus enables high efficiency and strong performance for long-form speech, the model has not been assessed on broader auditory

scenes (e.g., environmental sounds or music). Extending FastSLM toward general audio understanding remains an important direction for future work.

To address this limitation, we plan to extend training and evaluation to audio understanding tasks, enabling a more comprehensive assessment of the FastSLM ability to preserve detailed acoustic information.

