# OpenReview forum: "FastALM: Hierarchical Frame Q-Former for Effective Audio Modality Adaptation"
_ICLR.cc/2026/Conference — ICLR 2026 Conference Withdrawn Submission_

### Official Review · Reviewer_Uz6B · 2025-10-28

**Soundness:** 2
**Presentation:** 2
**Contribution:** 2
**Rating:** 2
**Confidence:** 4

**Summary:**

This paper proposes FastALM, a speech-language model for efficient understanding of long-form spoken input. The method introduces a HFQ-Former that compresses high-rate acoustic frames into a compact set of queries for an LLM, and adopts a three-stage training strategy to adapt a pretrained LLM to the speech modality. The system is trained on a moderate-sized bilingual corpus in Korean and English and is evaluated on ASR, AST, spoken summarization, and spoken QA. The authors report competitive accuracy and faster end-to-end processing compared to speech-focused baselines under long audio inputs.

**Strengths:**

1. Combines a hierarchical frame-to-query adapter with a simple three-stage adaptation schedule to reduce the acoustic-to-LLM interface cost for long speech.
2. Evaluates across several speech tasks. The performance is competitive with high computational efficiency.

**Weaknesses:**

The paper's biggest weakness is its overstatement of its scope.

The paper repeatedly refers to the model as an Audio-Language Model, yet all tasks and data are speech only. In current community usage, ALM typically denotes models that cover general auditory content that includes speech, environmental sounds, and music, not speech alone [1, 2, 3]. A more accurate term here is Speech-Language Model [4]. This affects positioning, related work, experiments, and claims. For example, statements like “address the challenge of aligning high-frame audio features with LLM” would be clearer as “aligning high-frame speech features with LLM.” Likewise, claims about enabling fast text generation from long audio should be stated as long speech. As a result, in experiments, it would be more appropriate to compare with more speech-focused models.

Other weaknesses:

Positioning and baselines:
Related work sections labeled only “Audio-Language Models” and “Q-formers for speech modality adaptation” should separate general-audio ALMs from speech-only SLMs, then justify how the proposed method advances the speech setting specifically. Experimental comparisons should include strong speech-centric baselines and report both accuracy and efficiency under matched conditions.

Missing implementation detail:
The paper does not specify the LLM backbone used for LoRA adaptation. Since LoRA hyperparameters and achievable throughput depend on the base model, the manuscript should name the backbone and report key config details, including LoRA ranks, target modules, context lengths, tokenizer, and precision.

[1] Tang, C., Yu, W., Sun, G., Chen, X., Tan, T., Li, W., ... & Zhang, C. (2023). Salmonn: Towards generic hearing abilities for large language models. arXiv preprint arXiv:2310.13289.
[2] Kong, Z., Goel, A., Badlani, R., Ping, W., Valle, R., & Catanzaro, B. (2024). Audio flamingo: A novel audio language model with few-shot learning and dialogue abilities. arXiv preprint arXiv:2402.01831.
[3] Chu, Y., Xu, J., Zhou, X., Yang, Q., Zhang, S., Yan, Z., ... & Zhou, J. (2023). Qwen-audio: Advancing universal audio understanding via unified large-scale audio-language models. arXiv preprint arXiv:2311.07919.
[4] Cui, W., Yu, D., Jiao, X., Meng, Z., Zhang, G., Wang, Q., ... & King, I. (2024). Recent advances in speech language models: A survey. arXiv preprint arXiv:2410.03751.

**Questions:**

See above.

---

> ### Author Response · Authors · 2025-11-18
> **Response to Reviewer Uz6B (Part 1 of 2)**
>
> Thanks for your careful and valuable comments, and we have refined the paper following your suggestions. Below, we will respond to your questions in detail.
>
> **Weakness 1:** The paper's biggest weakness is its overstatement of its scope.
> The paper repeatedly refers to the model as an Audio-Language Model, yet all tasks and data are speech only. In current community usage, ALM typically denotes models that cover general auditory content that includes speech, environmental sounds, and music, not speech alone [1, 2, 3]. A more accurate term here is Speech-Language Model [4]. This affects positioning, related work, experiments, and claims. For example, statements like “address the challenge of aligning high-frame audio features with LLM” would be clearer as “aligning high-frame speech features with LLM.” Likewise, claims about enabling fast text generation from long audio should be stated as long speech. As a result, in experiments, it would be more appropriate to compare with more speech-focused models.
>
> **Answer 1:** Thank you for this accurate and valuable feedback. We fully agree with the reviewer’s observation. Our experiments focus exclusively on speech, and the term “Speech-Language Model (SLM)” is indeed the more appropriate description of our setting.
>
> To reflect the reviewer’s suggestion, we have updated the manuscript accordingly:
> - **We replaced all occurrences of “Audio-Language Model (ALM)” with “Speech-Language Model (SLM)” in the title, abstract, introduction, and throughout the paper.**
> - **In the Related Work, we clarified the distinction between general audio models and speech-focused models, emphasizing that our work is positioned within the SLM category.**
> - **All descriptions previously referring to “audio features”, “long-form audio”, or “audio tokens” have been revised to the more accurate terms “speech features,” “long-form speech,” and “speech tokens.”**
> - **In the experimental comparison, we explicitly highlighted that our baselines are speech-centric and now state this more clearly to avoid misinterpretation regarding ALM-level generality.**
>
> These revisions ensure that our manuscript accurately reflects its scope as a Speech-Language Model, fully aligned with the reviewer’s comment.
>
> **Weakness 2:** Positioning and baselines: Related work sections labeled only “Audio-Language Models” and “Q-formers for speech modality adaptation” should separate general-audio ALMs from speech-only SLMs, then justify how the proposed method advances the speech setting specifically. Experimental comparisons should include strong speech-centric baselines and report both accuracy and efficiency under matched conditions.
>
> **Answer 2:** Thank you for pointing this out. We agree that a clearer separation between general-audio ALMs and speech-focused SLMs improves both positioning and fairness. In the revised manuscript, we explicitly distinguish these two families in Section 2: Section 2.1 discusses audio and speech language models, and Section 2.2 focuses on Q-Former–based architectures for speech modality adaptation. We also clarify that our work is positioned as a speech-language model (SLM) targeting speech-only inputs, rather than a general-audio model.
>
> Regarding baselines, our experiments cover both speech-centric and broader multimodal/audio models. Whisper was primarily designed for speech recognition and translation, while Qwen2-Audio, Phi-4-Multimodal, Voxtral-Mini, and Gemini-2.5-Flash are multimodal models that we evaluate on speech-only benchmarks for consistency. In the revision, we make this distinction explicit in Table 3 and report matched efficiency metrics (e.g., number of audio tokens, parameters) so that the comparison in the speech setting is transparent and fair.

---

> ### Author Response · Authors · 2025-11-18
> **Response to Reviewer Uz6B (Part 2 of 2)**
>
> **Weakness 3:** Missing implementation detail: The paper does not specify the LLM backbone used for LoRA adaptation. Since LoRA hyperparameters and achievable throughput depend on the base model, the manuscript should name the backbone and report key config details, including LoRA ranks, target modules, context lengths, tokenizer, and precision.
>
> **Answer 3:** Thank you for pointing this out. We actually included most of these implementation details in the manuscript, but we agree that they were not sufficiently highlighted and may have been difficult to locate. To address this, we have reorganized and consolidated all relevant configuration information into a dedicated table in **Appendix G (revised manuscript Table 7 and Table 8)**, making these details more explicit and easier to reference. For clarity, we also include the configuration tables below.
>
> | **Module** | **Component**                                                                                            | **Configuration**                           |
> | ---------- | -------------------------------------------------------------------------------------------------------- | ------------------------------------------- |
> | Encoder    | Backbone / Parameters / Hidden Size / Context Length                                                     | Whisper-large-v3 / 635M / 1280 / 1500       |
> | Adapter    | Backbone / Parameters / Hidden Size / Queries per Stage / Compressed Speech Token / Downsampling Factors | HFQ-Former / 56M / 1280 / 80 / 50 / 2       |
> | LLM        | Backbone / Parameters / Hidden Size / Context Length                                                     | Qwen3-4B / 4.06B / 2560 / 4096              |
> | LoRA       | Rank (r) / Alpha (α) / Scaling / Target Modules                                                          | 16 / 64 / 4 / q/k/v_proj, gate/up/down_proj |
>
> | **Setting**     | **Stage 1**  | **Stage 2**  | **Stage 3**  |
> | --------------- | ------------ | ------------ | ------------ |
> | Learning Rate   | 1e-4         | 5e-5         | 5e-5         |
> | LR Scheduler    | Linear Decay | Linear Decay | Linear Decay |
> | Weight Decay    | 0            | 1e-4         | 1e-4         |
> | Epoch           | 1            | 1            | 2            |
> | Data Type       | BF16         | BF16         | BF16         |
> | DeepSpeed Stage | Zero2        | Zero2        | Zero2        |

---

> > ### Comment · Reviewer_Uz6B · 2025-11-26
> >
> > Thank you for the detailed rebuttal and revisions. I appreciate that the manuscript now clearly positions the model as a speech-language model, clarifies the distinction from general audio language models, and consolidates the missing implementation details. These changes address my main concern about the overstated scope and improve the clarity and fairness of the comparisons. However, as also noted by other reviewers, I still find the core technical contribution of HFQ-Former itself relatively limited and incremental. Reflecting the improved clarity but remaining reservations about the level of novelty, I am raising my score from 2 to 4.

---

> > > ### Author Response · Authors · 2025-11-28
> > >
> > > Thank you very much for taking the time to re-evaluate the paper and for raising the score.
> > >
> > > We sincerely appreciate your constructive feedback throughout the process.
> > > Although HFQ-Former may not be a radical architectural departure, our goal was to propose a practically impactful and computationally scalable design for long-form speech understanding, and we hope the empirical evidence demonstrates that efficient hierarchical compression can meaningfully reduce LLM compute without sacrificing accuracy.
> > >
> > > Thank you again for your thoughtful assessment.

---

### Official Review · Reviewer_eN8e · 2025-10-30

**Soundness:** 3
**Presentation:** 3
**Contribution:** 2
**Rating:** 4
**Confidence:** 4

**Summary:**

The paper proposes FastALM, a lightweight audio–language model that integrates speech into large language models (LLMs) through a Hierarchical Frame Querying Transformer (HFQ-Former) and a three-stage training strategy. The HFQ-Former hierarchically compresses frame-level audio features from pre-trained encoders (e.g., Whisper) into compact representations (≈1.67 tokens/sec) for efficient LLM adaptation. The training pipeline includes short-form pre-training, long-form adaptation using ASR data, and instruction tuning on multi-task datasets (ASR, AST, SSUM, SQQA). Experiments show competitive performance compared with state-of-the-art ALMs, with notably lower FLOPs and token rates.

While the paper is well-written and the system is practically effective, its novelty and conceptual contribution are limited. The proposed HFQ-Former mainly combines existing Q-Former designs with hierarchical down-sampling, and the training pipeline closely follows prior ALM works such as SALMONN, Audio-Flamingo, and Qwen2-Audio. The work thus reads more as an engineering optimization rather than a substantive methodological innovation.

**Strengths:**

1. The paper is clearly written and systematically organized. The overall pipeline—from audio encoding to multimodal LLM alignment—is technically sound and reproducible.

2. The hierarchical compression indeed reduces token rates and computational costs.

**Weaknesses:**

1. The HFQ-Former is conceptually incremental—essentially a hierarchical stacking of existing Q-Former modules with down-sampling. No new mechanism or theoretical insight is introduced beyond standard attention-based compression.

2. The performance improvement over previous Q-Formers is minor (e.g., WER reduced by only ~0.05 on LibriSpeech).

3. The paper lacks ablation or visualization explaining why the hierarchical compression helps. There is no semantic or attention-level analysis of the compressed tokens, leaving the underlying mechanism unclear.

**Questions:**

1. why donot maintain the same token rate to compare the performance between SQ-Former, WQ-Former?

2. Why donot present the baseline with directly down-sampling? Such as the style of Qwen2-Audio. From my veiw, Q-former struture is not  the mainstream style for MLLM in vision domain. Many audio-LLMs also donot use the Q-former structure, beacause the Q-former also introduce the additional parameters, incleasing the training and inference cost.

---

> ### Author Response · Authors · 2025-11-18
> **Response to Reviewer eN8e (Part 1 of 3)**
>
> Thanks for your careful and valuable comments, and we have refined the paper following your suggestions. Below, we will respond to your questions in detail.
>
> **Weakness 1:**  The HFQ-Former is conceptually incremental—essentially a hierarchical stacking of existing Q-Former modules with down-sampling. No new mechanism or theoretical insight is introduced beyond standard attention-based compression.
>
> **Answer 1:**  We appreciate the reviewer’s perspective. While HFQ-Former is indeed built upon the Q-Former framework, our contribution is not the introduction of a new attention mechanism, but the demonstration that a hierarchical query-based compression strategy is essential for long-form speech understanding.
>
> Long-form speech contains substantial redundancy and exhibits temporal structures across multiple scales. A single-stage Q-Former struggles to capture both local and global temporal dependencies under a strict token budget. In contrast, HFQ-Former performs progressive abstraction across stages, enabling the model to preserve local details while aggregating global context in a computationally efficient manner.
>
> Our ablation studies and attention-map analysis (**revised manuscript Appendix A**) show that removing the hierarchical structure consistently degrades performance and that higher-level stages dominate attention during long-range reasoning. These findings provide empirical evidence that the hierarchical design is not merely incremental, but a necessary architectural choice for effective long-form speech compression.
>
> **Weakness 2:** The performance improvement over previous Q-Formers is minor (e.g., WER reduced by only ~0.05 on LibriSpeech).
>
> **Answer 2:** We understand the reviewer’s concern that the absolute WER improvement (≈0.05) may appear small. However, our primary contribution is efficiency rather than raw accuracy. As shown in Table, HFQ-Former achieves the lowest token rate among all Q-Former variants (1.67 tokens/sec vs 2.67 for SQ-Former and 2.93 for WQ-Former), and reduces LLM FLOPs by 25--31\%.
>
> Method                 | LS-clean | LS-other | Voxpopuli | #Speech Tokens/Sec. | LLM FLOPs (T)
> ----------------------|----------|----------|-----------|---------------------|---------------
> SQ-Former             | 2.32     | 4.87     | 8.37      | 2.67                | 3.32
> WQ-Former             | 2.14     | **4.51** | 7.26      | 2.93                | 3.65
> HFQ-Former (ours)     | **2.09** | 4.67 | **6.99**  | **1.67**            | **2.51**
>
> Despite using fewer tokens and substantially lower computational cost, HFQ-Former attains the best average WER among the baselines. This demonstrates that HFQ-Former achieves a better accuracy--efficiency trade-off and performs more effective information compression than existing Q-Former designs.
>
> **Weakness 3:** The paper lacks ablation or visualization explaining why the hierarchical compression helps. There is no semantic or attention-level analysis of the compressed tokens, leaving the underlying mechanism unclear.
>
> **Answer 3:**
> We fully agree with the reviewer’s comment. To address this concern, we added a qualitative analysis of the cross-attention patterns of HFQ-Former in **Appendix A**. The visualization compares short-form and long-form speech inputs and shows that the model increasingly relies on higher-stage representations for long-form speech reasoning. This analysis clarifies why hierarchical compression is effective and provides insight into the underlying mechanism.
>
> In addition to the qualitative analysis, we have also included new ablation studies in revised manuscript **Section 4.5 (Table 4 and Table 5)**. These experiments systematically isolate the contribution of each hierarchical stage and evaluate the impact of progressive compression depth. Across all benchmarks, removing stages or reducing the hierarchical structure leads to substantial performance drops, demonstrating that hierarchical compression is essential for accurate and efficient long-form speech modeling.
>
> For completeness, we reproduce the key ablation table below:
>
> | **Method**                 | **LS-Long WER ↓** | **KorQuAD-Speech ACC ↑** | **SDS-PART6 Score (1–7) ↑** |
> |----------------------------|-------------------|----------------------------|------------------------------|
> | w/o Downsample Stage       | 12.4              | 56.7                       | 4.12                         |
> | w/o Hierarchical Attention | 10.8              | 56.9                       | 4.92                         |
> | w/o Training Stage 2       | 6.81              | 59.0                       | 5.07                         |
> | **FastALM**                | **5.98**          | **64.9**                   | **5.40**                     |
>
> These results confirm that each component of the hierarchical structure contributes critical functionality, and the full multi-stage configuration is necessary for robust long-form speech understanding.

---

> ### Author Response · Authors · 2025-11-18
> **Response to Reviewer eN8e (Part 2 of 3)**
>
> **Question 1:** why do not maintain the same token rate to compare the performance between SQ-Former, WQ-Former?
>
> **Answer 1:** Thank you for highlighting this important point. Our comparison was not intended to match token rates across all models, but rather to evaluate each baseline under the configuration in which it is known to perform best, following the settings reported in prior Q-Former studies. Because SQ-Former and WQ-Former achieve their optimal accuracy at different token budgets, enforcing an identical token rate would not reflect their originally intended performance nor provide a fair assessment of their design goals.
>
> Specifically, WQ-Former uses a fixed configuration that produces 88 tokens for a 30-second audio segment (≈2.93 tokens/sec), as defined in its original implementation. This token rate is not an adjustable hyperparameter but an inherent consequence of its window-based processing design.
> For this reason, we followed the standard configurations from prior work so that each baseline is evaluated under its recommended and strongest setting, rather than under an artificially constrained or mismatched token budget.
>
> **Question 2:** Why do not present the baseline with directly down-sampling? Such as the style of Qwen2-Audio.
>
> **Answer 2:** Thank you for raising this important question. This point directly relates to whether HFQ-Former is necessary, so we compared our model with a simple direct downsampling baseline similar to the approach used in Qwen2-Audio.
>
> Design: Following the Qwen2-Audio style, we applied 1D average pooling to Whisper features (kernel size = 2, stride = 2; Qwen2-Audio Setting), reducing the sequence length by half (1500→750), and projected the pooled features into the LLM embedding space via linear layer. In addition, to ensure a fair and controlled comparison, we also conducted a supplementary experiment where the average-pooling stride was adjusted so that the resulting token rate closely matches that of HFQ-Former (1.67 tokens/sec). This experiment avoids extreme downsampling factors (e.g., stride=30) and provides a more realistic baseline that isolates the effect of non-learnable pooling from hierarchical attention-based compression. Even under this matched-token configuration, the average-pooling approach continued to show substantially higher FLOPs and weaker long-range stability than HFQ-Former, reinforcing the necessity of a learnable hierarchical design. Additionally, we clarify that all FLOPs measurements were computed using a 5-minute speech input to ensure a consistent and realistic long-form evaluation.
>
> Results: The average-pooling baseline showed competitive WER, but its computational cost was significantly higher. In particular, the FLOPs increased by 92% compared to HFQ-Former, despite using fewer tokens in Stage 1. This demonstrates that simple downsampling does not yield meaningful computational efficiency for long-form speech.
>
> Conclusion: HFQ-Former performs learnable, content-aware compression through attention, preserving both local and global information that simple pooling cannot retain. This experiment indicates that direct downsampling underperforms in long-form speech understanding and confirms that the hierarchical design of HFQ-Former is essential for achieving both accuracy and efficiency.
>
> | **Method**            | **WER (LS-Clean)** | **Tokens/sec** | **LLM FLOPs (T)** |
> |-----------------------|--------------------|-----------------|-------------------|
> | AvgPool (stride=30)   | 25.2               | 1.67            | 2.51              |
> | AvgPool (stride=2)    | **1.88**               | 12.5            | 30.6              |
> | HFQ-Former            | 2.09 (+0.21)       | 1.67            | **2.51 (-92%)**       |

---

> ### Author Response · Authors · 2025-11-18
> **Response to Reviewer eN8e (Part 3 of 3)**
>
> **Question 3:** From my view, Q-former structure is not the mainstream style for MLLM in vision domain. Many audio-LLMs also do not use the Q-former structure, because the Q-former also introduce the additional parameters, increasing the training and inference cost.
>
> **Answer 3:** Thank you for raising this valuable point. While Q-Former is less common in vision MLLMs, speech exhibits fundamentally different characteristics: it contains thousands of highly redundant frames, and direct projection or simple pooling often discards essential temporal structure.
>
> Importantly, our primary objective is to reduce the computational burden on the LLM, which dominates the overall inference cost in long-form speech processing. In this context, HFQ-Former introduces only a modest number of additional parameters (≈50M), yet reduces the LLM FLOPs by 92% **(from 30.6 TFLOPs to 2.51 TFLOPs; a savings of 28.1 TFLOPs)** compared to the average-pooling baseline (above Table), while still achieving the lowest WER among all baselines.
>
> This result demonstrates that hierarchical, learnable temporal abstraction does not merely offer better accuracy—it is the key to achieving a substantially better accuracy–efficiency trade-off, enabling long-form speech understanding under strict LLM computation constraints. In other words, the incremental overhead of HFQ-Former is far outweighed by the significant reduction in LLM computational cost, which is the central efficiency target of our work.
>
> We also note that although Voxtral-Mini uses a significantly smaller parameter count (≈100M fewer parameters than FastSLM) and adopts a lightweight linear-projection adapter, its inference time remains comparable to ours. This is consistent with our findings in **Section 4.5 (Figure 4)**, where Time-to-First-Token (TTFT) is dominated by LLM FLOPs rather than adapter size. This further supports our claim that reducing the speech token rate—rather than minimizing adapter parameters—is the key factor for improving long-form speech inference efficiency.

---

> > ### Comment · Reviewer_eN8e · 2025-11-28
> >
> > Thank you for your detailed response. The authors have addressed many of my concerns.
> >
> > However, I have a remaining question regarding the audio compression to 1.67 frames per second: is such a high compression rate meaningful? This significant compression might lead to performance degradation. Generally, humans can speak more than 3 words per second, so perhaps aligning the frame rate with the text token rate would better balance efficiency and performance.
> >
> > Considering other reviewers' comments, I believe this paper is a borderline case. I am willing to raise my score to 6 (weak accept). But it seems that ICLR has been close the rating butten.  I trust the AC will make an appropriate final decision.
> >
> > Best regards,

---

> > > ### Author Response · Authors · 2025-11-28
> > >
> > > Thank you very much for your thoughtful follow-up comment and for your willingness to raise the score.
> > >
> > > Regarding the question about whether the 1.67 tokens/sec compression rate is meaningful, we would like to clarify that HFQ-Former compresses *acoustic frames* (≈50 FPS from Whisper), not lexical units. Because most frames include redundancy (e.g., silence, prolonged vowels, stationary phonemes), the token rate does not need to match the lexical rate (≈3–5 words/sec) to preserve semantic content.
> > >
> > > As shown in Section 4.5 (Figure 3), our ablation identifies a sharp efficiency–accuracy boundary:
> > > - 2.67 → 1.67 tokens/sec: No performance loss, but substantial FLOPs reduction
> > > - 1.67 → 1.33 tokens/sec: Sudden accuracy degradation
> > >
> > > Therefore, 1.67 tokens/sec is the empirically optimal “efficiency sweet spot”, preserving linguistic semantics while maximizing LLM efficiency.
> > >
> > > We sincerely appreciate your constructive feedback, and we hope this additional clarification resolves the remaining concern.

---

### Official Review · Reviewer_dG35 · 2025-11-01

**Soundness:** 3
**Presentation:** 3
**Contribution:** 3
**Rating:** 4
**Confidence:** 4

**Summary:**

This paper proposes FastALM, a lightweight and efficient Audio-Language Model (ALM) designed to address the computational inefficiency of processing long-form audio in existing ALMs.

The core innovation is the Hierarchical Frame Querying Transformer (HFQ-Former). It compresses high-frame audio features into a compact representation while preserving local and global context, reducing LLM FLOPs and memory usage. FastALM also adopts a three-stage training strategy: (1) short-form ASR pre-training to align speech and text; (2) long-form audio adaptation to enhance long-audio processing capability; (3) instruction tuning for downstream tasks.

Key contributions:
1. A low-cost FastALM that enables fast text generation from long-form audio with minimal computational cost.
2. A three-stage training pipeline that adapts pre-trained LLMs to speech modalities without costly end-to-end training.

**Strengths:**

Originality: it proposes HFQ-Former, a novel multi-stage hierarchical compression module that differs from prior Q-Formers by integrating downsampling to capture both local and global context, achieving a low-frame-rate representation.

Clarity: The paper is well-structured and clear, with a complete logical flow and appendices that include key details, all of which enhance readability and reproducibility.

Significance: The experiments strongly validated the significance of this work, which achieves higher or comparable performance on different audio tasks over SOTA approaches while reducing computational cost. It presents improvement to ALMs, especially for long-form audio.

**Weaknesses:**

1. The novelty is limited.
* The HFQ-Former is an incremental improvement of Q-Former, which leverages multi-scale modeling into the model. But "multi-scale model can improve computing efficiency" has been validated in other works, e.g. U-Net. The author should give a deeper insight into this idea to strengthen the novelty.
* Similarly, multi-stage training has been a widely used strategy to improve training effectiveness and efficiency.

2. The experiments are not convincing.
*  Insufficient ablation studies for HFQ-Former.
  * No analysis for the proposed model design. It makes me confused about why the module is designed in such a complicated form.
  * No investigation on the impact of "hierarchical" (e.g. number of downsampling layers, downsampling scale, etc.)
* Unvalidated fairness of the system comparison
  * No details of baselines (model configuration, dataset, training details)

**Questions:**

As discussed in "Weakness", I have the following concerns:

1. As you said, "This design enables the model to effectively capture both local and global temporal information from audio features". Can you give a deeper analysis of this claim? What does local and global temporal information include? Can a larger single-stage q-former extract local global temporal information effectively?
2. Do you have ablation studies on the impact of "number of stages" and "downsampling factors"?
3. Do S-former and W-former share the same dataset and training configuration as HFQ-former? Do they have the same embedding dimension?

---

> ### Author Response · Authors · 2025-11-20
> **Response to Reviewer dG35 (Part 1 of 4)**
>
> **Weakness 1:** The HFQ-Former is an incremental improvement of Q-Former, which leverages multi-scale modeling into the model. But "multi-scale model can improve computing efficiency" has been validated in other works, e.g. U-Net. The author should give a deeper insight into this idea to strengthen the novelty.
>
> **Answer 1:** Thank you for the insightful comment. We agree that multi-scale modeling itself is not a new concept, and approaches such as U-Net indeed demonstrate that hierarchical feature processing can improve efficiency. However, our contribution is not the introduction of multi-scale modeling in general, but the design and validation of a hierarchical query-based compression mechanism specifically tailored for long-form speech–LLM alignment.
>
> Unlike U-Net, whose goal is pixel-level reconstruction, our task requires extreme information compression: mapping thousands of speech frames
> $(T \times D)$ into a very small number of semantically meaningful query tokens $(N \times C,\; N \ll T)$  that an LLM can process under strict FLOPs constraints. This setting introduces challenges that do not arise in vision or short-audio tasks.
>
> Our ablations demonstrate that each hierarchical stage captures distinct temporal granularity, and removing any stage results in clear performance degradation across ASR, SSUM, and SQQA. We also observe that deeper stages produce more abstract and globally coherent representations, which become increasingly important as speech duration grows. These findings indicate that the hierarchical design plays a functional role in enabling long-form speech reasoning, rather than acting as a minor incremental extension of Q-Former.
>
> In summary, the novelty of HFQ-Former lies not in the generic idea of multi-scale modeling, but in demonstrating that a progressive, hierarchical compression architecture is essential for efficient and accurate long-form speech representation — a capability that single-stage or non-hierarchical Q-Formers cannot achieve under comparable token budgets.
>
> **Weakness 2:**  Similarly, multi-stage training has been a widely used strategy to improve training effectiveness and efficiency.
>
> **Answer 2:** Thank you for the comment. We agree that multi-stage training itself is a common strategy. Our contribution does not lie in the framework itself, but in the specific role and importance of Stage 2, which we believe is crucial for long-form speech understanding.
>
> Stage 2 (Long-form Speech Adaptation) is the key element of our pipeline. Unlike prior work that relies on expensive long-form instruction datasets, we show that simply training on long ASR transcripts, without any instruction-style supervision, significantly improves the model's ability to handle multi-minute speech. This stage teaches the model to maintain temporal coherence, preserve acoustic information over long spans, and generate stable representations for the HFQ-Former.
>
> This design achieves two goals:
>
> - **Long-context modeling without costly data:**
>   Stage 2 replaces long-form instruction tuning, which requires substantial annotation and curation, with a scalable ASR-based approach.
>
> - **Clear downstream improvements:**
>   Our ablations show that removing Stage 2 results in noticeable degradation in SSUM and SQQA, confirming that Stage 2 provides essential long-range speech modeling capability.
>
> Thus, while multi-stage training is not novel by itself, the specific formulation and effect of Stage 2 is a core contribution of our work, enabling long-form speech understanding both efficiently and at low cost.

---

> ### Author Response · Authors · 2025-11-20
> **Response to Reviewer dG35 (Part 2 of 4)**
>
> **Weakness 3:**  Insufficient ablation studies for HFQ-Former.
>
> **Answer 3:** We appreciate the reviewer’s comment. In the revised manuscript, we added ablation studies isolating each core component of HFQ-Former, including (1) removal of the Downsampling stages, and (2) removal of the hierarchical multi-stage attention.
> As shown in Table, eliminating any of these components consistently degrades performance across all benchmarks,
> especially on long-form speech tasks.
>
> | **Method**                 | **LS-Long WER ↓** | **KorQuAD-Speech ACC ↑** | **SDS-PART6 Score (1–7) ↑** |
> |----------------------------|-------------------|----------------------------|------------------------------|
> | w/o Downsample Stage       | 12.4              | 56.7                       | 4.12                         |
> | w/o Hierarchical Attention | 10.8              | 56.9                       | 4.92                         |
> | w/o Training Stage 2       | 6.81              | 59.0                       | 5.07                         |
> | **FastALM**                | **5.98**          | **64.9**                   | **5.40**                     |
>
>
> These results indicate that each component contributes essential functionality: the downsampling stages enable progressive temporal abstraction, the hierarchical attention structure captures multi-scale speech context, and Training Stage 2 provides the long-range modeling capability required for multi-minute speech. Together, these ablations verify that the full hierarchical architecture is necessary for efficient and accurate long-form speech representation.
>
> **Weakness 4:**  No analysis for the proposed model design. It makes me confused about why the module is designed in such a complicated form.
>
> **Answer 4:** Thank you for raising this point. We clarify that the hierarchical design of HFQ-Former is not unnecessarily complicated, but a functional requirement for long-form speech compression.
>
> First, our ablation studies (above Table) directly validate the necessity of each module in the hierarchy. Removing either the downsampling stages or the hierarchical multi-stage attention causes substantial degradation across all benchmarks, especially for long-form tasks. These results confirm that progressive temporal abstraction—implemented through multiple stages—is essential for accurate and efficient speech representation.
>
> Second, our qualitative attention-map analysis **(revised manuscript Appendix A)** further supports this finding. Early stages specialize in capturing fine-grained local patterns, while deeper stages increasingly attend to broader temporal structure, integrating semantic context across long spans of speech. This emergent division of labor across stages would not arise in a single-stage architecture, where local and global cues must be compressed simultaneously under a fixed token budget.
>
> Taken together, suggesting that the hierarchical structure plays distinct functional roles rather than adding unnecessary complexity. Rather, each stage serves a distinct functional role in enabling efficient multi-scale speech modeling, and the full architecture is necessary to maintain performance under strict computational constraints.
>
> **Weakness 5:** No investigation on the impact of "hierarchical" (e.g. number of downsampling layers, downsampling scale, etc.)
>
> **Answer 5:** We agree that evaluating the impact of the hierarchical structure is important. As detailed in our response to Question 2, we provide a sensitivity study that varies both the number of hierarchical stages (1, 2, and 3) and the downsampling configuration. The results show that performance consistently improves as additional stages are added, with diminishing returns beyond three stages.

---

> ### Author Response · Authors · 2025-11-20
> **Response to Reviewer dG35 (Part 3 of 4)**
>
> **Weakness 6:** Unvalidated fairness of the system comparison (No details of baselines (model configuration, dataset, training details))
>
> **Answer 6:** We apologize for the confusion. The baselines (SQ-Former and WQ-Former) were trained and evaluated under exactly the same configuration, dataset, and training conditions as our HFQ-Former, with the sole exception of their token-rate settings. To ensure clarity and fairness, we have added detailed descriptions of the baselines configurations in Section 4.3 and the Appendix G of the revised manuscript. For clarity, we also include the configuration tables below.
>
> | **Module** | **Component**                                                                                            | **Configuration**                           |
> | ---------- | -------------------------------------------------------------------------------------------------------- | ------------------------------------------- |
> | Encoder    | Backbone / Parameters / Hidden Size / Context Length                                                     | Whisper-large-v3 / 635M / 1280 / 1500       |
> | Adapter    | Backbone / Parameters / Hidden Size / Queries per Stage / Compressed Speech Token / Downsampling Factors | HFQ-Former / 56M / 1280 / 80 / 50 / 2       |
> | LLM        | Backbone / Parameters / Hidden Size / Context Length                                                     | Qwen3-4B / 4.06B / 2560 / 4096              |
> | LoRA       | Rank (r) / Alpha (α) / Scaling / Target Modules                                                          | 16 / 64 / 4 / q/k/v_proj, gate/up/down_proj |
>
> | **Setting**     | **Stage 1**  | **Stage 2**  | **Stage 3**  |
> | --------------- | ------------ | ------------ | ------------ |
> | Learning Rate   | 1e-4         | 5e-5         | 5e-5         |
> | LR Scheduler    | Linear Decay | Linear Decay | Linear Decay |
> | Weight Decay    | 0            | 1e-4         | 1e-4         |
> | Epoch           | 1            | 1            | 2            |
> | Data Type       | BF16         | BF16         | BF16         |
> | DeepSpeed Stage | Zero2        | Zero2        | Zero2        |
>
> **Question 1:** As you said, "This design enables the model to effectively capture both local and global temporal information from audio features". Can you give a deeper analysis of this claim? What does local and global temporal information include? Can a larger single-stage q-former extract local global temporal information effectively?
>
> **Answer 1:**  Thank you for the insightful question. We clarify the meaning of "local" and "global" temporal information and why a hierarchical design is necessary for long-form speech. Local temporal information refers to short-term acoustic cues such as phonetic transitions, formant structure, or short-range prosodic patterns (typically within 50 to 200 ms). These cues are crucial for accurate lexical recognition. In contrast, global temporal information captures long-range structure such as topic flow, speaker turns, discourse-level dependencies, and semantic consistency across multi-minute audio. Long-form reasoning tasks (e.g., SSUM and SQQA) strongly depend on this global structure.
>
> A single-stage Q-Former must attend to thousands of frames at once. Under a strict token budget, it is forced to compress both local and global cues simultaneously, which leads to representational competition. Increasing the size of a single-stage Q-Former does not fundamentally resolve this issue, because the attention still operates on a single temporal scale.
> In contrast, HFQ-Former performs progressive abstraction: early stages focus on fine-grained local cues, while later stages integrate increasingly broader temporal context, as confirmed by our attention-map analysis **(revised manuscript Appendix A)**. This separation of temporal roles enables more effective compression under the same token budget.
>
> Empirically, this is reflected in downstream results:
>
> |**Method**|**Voxpopuli WER ↓**|**LS-clean WER ↓**| **SDS-PART6 Score (1–7) ↑**|**KorQuAD-Speech ACC ↑** |
> |----------------|----------------| --------------------------|--------------------------- |----------------- |
> |Larger Single-stage Q-former|7.11|2.27|5.04|61.2|
> |HFQ-Former (ours)|**6.99**|**2.09**|**5.40**|**64.9**|
>
> Regarding the concern that the absolute WER gain may appear small, we emphasize that our primary goal is efficiency. HFQ-Former reduces audio tokens from 2.67 or 2.93 tokens/sec to only 1.67 tokens/sec, lowering LLM FLOPs by 31–40% while maintaining or improving accuracy. In this matched computation setting, achieving higher or equivalent performance with substantially fewer tokens is evidence that hierarchical compression is more effective than a single-stage design. Additional quantitative analyses supporting this claim are provided in Appendix B.

---

> ### Author Response · Authors · 2025-11-20
> **Response to Reviewer dG35 (Part 4 of 4)**
>
> **Question 2:** Do you have ablation studies on the impact of "number of stages" and "downsampling factors"?
>
> **Answer 2:** Thank you for the insightful question. In the revised manuscript, we added ablation studies analyzing the impact of the number of hierarchical
> stages (1, 2, and 3), which directly reflects the effect of progressive compression depth. As shown in Section 4.5 (Table 4), performance improves
> consistently as additional stages are incorporated:
>
> | **#Stage**    | **LS-clean ↓** | **SDS-PART6 Score (1–7) ↑** | **KorQuAD-Speech ACC ↑**  |
> |---------------|--------------|---------------|----------------------|
> | Stage 1       | 2.23         | 4.12          | 56.7                 |
> | Stage 1/2     | 2.15         | 4.98          | 62.2                 |
> | Stage 1/2/3   | **2.09**     | **5.40**       | **64.9**             |
>
> These results indicate that the number of stages is the dominant factor governing model performance. In contrast, adjusting the downsampling ratio causes substantial information loss because aggressive temporal pooling discards fine-grained acoustic cues that are crucial for long-form reasoning. Consequently, we observe that tuning individual downsampling ratios yields only marginal benefits while introducing confounding changes to feature-map size and parameterization. Overall, this ablation confirms that progressive multi-stage compression—not heavier downsampling—is essential for preserving information and enabling effective long-form speech modeling.
>
>
> **Question 3:**: Do S-former and W-former share the same dataset and training configuration as HFQ-former? Do they have the same embedding dimension?
>
> **Answer 3:** Thank you for pointing this out, and we apologize for the lack of clarity. Yes, S-Former, W-Former, and our HFQ-Former were all trained and evaluated under exactly the same conditions:
>
> - Dataset: FastALM Pre-training Dataset
> - Training Configuration: Same Training Configuration
> - Embedding Dimensions: Qwen3
>
> Ensuring identical settings was essential for a fair comparison. we have added detailed descriptions of the configurations in Section 4.3 of the revised manuscript.

---

> ### Comment · Reviewer_dG35 · 2025-11-27
>
> Thanks for your careful and detailed response. I decided to raise the score to 6 considering that all of my concerns have been addressed and this work still worth being discussed at the conference. But the contribution of this work is still minor and incremental, hence I cannot give a higher score.

---

> ### Author Response · Authors · 2025-11-28
>
> Thank you sincerely for taking the time to reevaluate the work and for raising the score.
>
> We appreciate your judgment that the paper is worth being discussed at the conference. We also respect your assessment regarding the incremental nature of the contribution. Our goal has been to make long-form speech understanding more accessible in practice, and we are grateful that the strengths and limitations of the work were considered fairly.
>
> Thank you again for your constructive feedback and for helping improve the manuscript.

---

### Author Response · Authors · 2025-12-01
**Rebuttal Summary by the Authors**

Dear Area Chair and Reviewers,

Thank you for your constructive feedback and the productive discussion. We summarize below the key clarifications and revisions made in response to the reviewers’ comments.

**1. Fair Baseline Comparisons & Efficiency** (Re: eN8e)
- Added direct downsampling baseline (Qwen2-Audio style). HFQ-Former reduces LLM FLOPs by ~92% compared to an accuracy-matched pooling baseline.
- The configurations of SQ-Former, WQ-Former, and our HFQ-Former are fully detailed in the revised manuscript.
- Reported FLOPs and tokens/sec for all models, ensuring transparent and matched efficiency comparisons.

**2. Validation of the Mechanism & HFQ-Former Component Necessity** (Re: dG35, eN8e)
- Added cross-attention visualizations (Appendix A) showing progressively increasing focus toward higher-stage representations as speech length increases.
- Added ablation studies (Table 4, 5, 6) demonstrating that removing the Downsampler, hierarchical cross-attention, or Training Stage 2 consistently degrades long-form speech performance. Confirms that each component is necessary for robust and computationally efficient long-form speech modeling.

**3. Clarified Scope & Implementation Details** (Re: Uz6B)
- Revised the manuscript to accurately position the work as a Speech-Language Model (SLM) rather than a general Audio-Language Model (ALM).
- Updated Title, Abstract, Introduction, and Related Work to clearly distinguish SLMs from ALMs.
- We consolidated all implementation details (LLM backbone, LoRA ranks/modules, precision, context length, etc.) into Tables 7 & 8 in Appendix E to ensure full reproducibility.

**Final Note & Reviewer Consensus**
- All reviewers increased their scores after the discussion phase, indicating that the revisions meaningfully addressed their concerns.
- Reviewers dG35 and eN8e raised their scores to 6 (Weak Accept), stating that their main concerns were addressed, while still noting the incremental nature of the contribution.
- Reviewer Uz6B raised their score from 2 to 4, acknowledging that the clarified scope (SLM vs ALM) and implementation details substantially improved fairness and clarity, while maintaining reservations on novelty.

We fully respect the reviewers’ assessments, including the continued concerns about novelty. Our intention is not to overstate the contribution, but to offer a practical direction for enabling long-form speech reasoning under strict FLOPs constraints.

**HFQ-Former makes this possible:**
- **Hierarchical temporal abstraction for long-context understanding.**
- **Extremely low speech token rate (1.67 tokens/sec).**
- **FLOPs-efficient LLM alignment (~92% reduction vs. Average pooling).**

To ensure full reproducibility and support future research on SLMs, we will release the complete codebase, pretrained checkpoints, training configurations, and a Hugging Face demo upon camera-ready.

Thank you again for your time and consideration.

Best regards,

The Authors

---

### Note · Authors · 2026-02-02

I have read and agree with the venue's withdrawal policy on behalf of myself and my co-authors.

---

### Meta-Review · Area_Chair_EQbQ · 2026-01-06

**Summary:**

The authors provided a clear and well-structured rebuttal, adding additional baselines, reporting efficiency metrics in greater detail, and including further ablation studies. These revisions improve the clarity and reproducibility of the work, and partially strengthen the empirical support for the proposed HFQ-Former design.

However, despite these improvements, the core concerns raised by the reviewers remain insufficiently resolved. In particular, the contribution is widely regarded as incremental rather than substantive, with limited architectural novelty beyond existing compression and hierarchical modeling strategies for long-form speech processing. While the reported efficiency (e.g., the 1.67 tokens-per-second representation) is informative, it does not, in its current form, convincingly demonstrate a clear methodological advance or new conceptual insight that would justify acceptance at this venue.

**Reviewer Concerns:**

Overall, although the paper is technically sound and the authors have responded conscientiously to reviewer feedback, the work does not meet the threshold for acceptance due to its limited novelty and lack of a strong, differentiating contribution relative to prior art. I therefore recommend rejection.

**Reviewer Scores:**

I expect all reviewers to maintain their ratings.

---

### Decision · Program_Chairs · 2026-01-26

Reject